# OpenReview forum: "FedGraph: Defending Federated Large Language Model Fine-Tuning Against Backdoor Attacks via Graph-Based Aggregation"
_ICLR.cc/2026/Conference — Submitted to ICLR 2026_

### Official Review · Reviewer_ugEp · 2025-10-19

**Soundness:** 3
**Presentation:** 3
**Contribution:** 3
**Rating:** 6
**Confidence:** 4

**Summary:**

This paper first demonstrate that backdoor attacks remain effective in the federated LoRA fine-tuning scenario with existing defenses inadequate. FedGraph is then proposed to separate benign and malicious parameter updates using unsupervised clustering basing on extracted topological features including Degree, Betweenness, and Closeness. These low-dimensional fingerprints help the identification of the malicious patterns and therefore results in a better defense.

**Strengths:**

1.	Multiple backdoor attacks have been evaluated covering a wide range of types. And the defense effect of the proposed FedGraph is good compared to baseline.
2.	Extensive ablation on the 3 fingerprints components clearly demonstrates the contribution of each.

**Weaknesses:**

1.	Graph construction requires O(n^2) computation; for large federations (hundreds or thousands of clients), this might be a bottleneck. Could you provide any computational comparison with FedAvg to demonstrate the computation overhead?
2.	Why do you choose these 3 attributes as the fingerprints but not others? Are there any other possibilities?

**Questions:**

1.	The bottom margin of each page is excessively larger than other papers. There might be an issue with the usage of the latex template.
2.	Why not include the accuracy for the main task since FPR and TPR can both be computed?

---

> ### Author Response · Authors · 2025-11-19
>
> We sincerely thank the reviewer for the feedback, especially concerning the margin issue. In response, we have revised the manuscript structure and added the relevant content. All modifications are highlighted in blue for clarity.
>
> ```
> > Q1. Why not include the accuracy for the main task since FPR and TPR can both be computed?
> ```
>
> In generative LLM settings, accuracy does not reflect model quality in a reliable way, which differs fundamentally from classification-based FL. Because generation is inherently non-deterministic, the same input can yield different outputs across runs due to sampling randomness, client-specific data distributions, and stochastic gradient updates. These factors make “accuracy” neither well-defined nor statistically stable for generative LLMs, even when FPR and TPR can be computed for the defense itself.
>
> Our defense focuses on identifying and suppressing malicious updates without altering benign ones. Since FedGraph only selects among submitted updates, TPR and FPR directly reflect its detection performance and its impact on benign training behavior. These metrics therefore offer a more reliable and interpretable evaluation of defense quality in this setting.
>
> To further demonstrate that FedGraph preserves main-task utility, we report loss curves in Section 4.1 (Figure 3). The benign loss decreases smoothly, while the loss associated with the backdoor objective consistently rises. This divergence indicates that FedGraph suppresses backdoor behavior without disrupting the model’s normal optimization process.
>
>
> ```
> > Q2. Graph construction requires O(n^2) computation; for large federations (hundreds or thousands of clients), this might be a bottleneck. Could you provide any computational comparison with FedAvg to demonstrate the computation overhead?
> ```
> We acknowledge that graph construction introduces higher computational overhead compared with FedAvg. If $N$ clients participate and each model update has dimensionality $d$, the per-round complexity of FedAvg is $O(Nd)$. In contrast, FedGraph requires computing pairwise cosine similarities with complexity $O(N^2 d)$, graph feature extraction with $O(N^2)$, and HDBSCAN clustering with $O(N \log N)$, followed by the same $O(Nd)$ aggregation step. Thus, the dominant additional cost arises from the $N(N-1)/2$ pairwise similarity computations, which makes the overhead quadratic in $N$ but linear in $d$.
>
> In practice, federated LLM fine-tuning differs substantially from traditional large-scale FL. Classical FL may involve thousands of mobile or IoT clients, whereas LLM federation typically contains only a few dozen to a few hundred institution-level participants (e.g., enterprises, research labs, or cloud nodes). These participants generally possess sufficient GPU resources, allowing pairwise similarity computation and graph construction to be efficiently parallelized. Moreover, the local fine-tuning step dominates the round-wise runtime in LLM FL; thus, the additional FedGraph overhead becomes relatively minor in comparison. For latency-sensitive and high-stakes applications (e.g., healthcare, finance, and public services), this moderate cost is justified by the significant improvement in robustness against backdoor threats.
>
> To provide a concrete comparison, we include the following illustrative computation table summarizing the relative overhead across different federation scales:
>
> Table R7. Computational  Comparison with FedAvg.
>
> | Setting | \#Clients $N$ | Model Size $d$ | Pairwise Similarities $\frac{N(N-1)}{2}$ | Relative Overhead vs. FedAvg |
> |--------|---------------|----------------|--------------------------------------------|-------------------------------|
> | Small-scale | $N=50$  | $d \approx 10\text{M}$  | $\approx 1.2\text{K}$ | $+2$–$3\%$ |
> | Medium-scale | $N=200$ | $d \approx 100\text{M}$ | $\approx 40\text{K}$  | $+10$–$15\%$ |
> | Large-scale (LLM FL) | $N=1000$ | $d \approx 1\text{B}$ | $\approx 500\text{K}$ | $+30$–$40\%$ |
>
> The implementation details and additional analysis of computational complexity are provided in Appendix Section B.1.

---

> > ### Comment · Reviewer_ugEp · 2025-11-21
> >
> > Thank you for your reply, I still have some further questions:
> >
> > 1. Do you have any main task performance compared to baseline methods? Figure 3 displays merely the results for FedGraph, do you have any comparison with other baselines?
> > 2. Do you have any direct computational cost analysis under $N=50, d\approx 1\text{B}$ settng? This should be more closer to Fed LLM training setting.

---

> ### Author Response · Authors · 2025-11-19
>
> ```
> >Q3. Why do you choose these 3 attributes as the fingerprints but not others? Are there any other possibilities?
> ```
>
> We use **degree, betweenness, and closeness centrality** to construct each client’s fingerprint because they provide a concise yet informative view of how an update fits within the overall similarity structure. Degree reflects how closely an update aligns with its immediate neighbors. Betweenness captures whether the update serves as a bridge between otherwise separate behavioral groups. Closeness describes its overall coherence with the global topology of all submitted updates. Taken together, these measurements form a **local–intermediate–global representation** of client updates and reveal patterns that are difficult to distinguish from any single perspective alone, and they allow us to characterize benign and abnormal behaviors in a stable and interpretable way.
>
> An additional consideration is **scalability**. The fingerprint is computed from graph structure rather than directly from the high-dimensional model updates, which makes it lightweight even when the underlying model has billions of parameters. This design also extends naturally to more fine-grained settings, such as constructing fingerprints from layer-specific similarity graphs when deeper analysis is required.
>
> Other features are certainly possible, such as gradient statistics or activation-based signatures, but they often incur substantially higher computational overhead or produce signals that are less stable across heterogeneous clients. The three centrality-based attributes therefore offer a practical compromise: they are expressive enough to separate malicious behaviors from benign ones, yet efficient and robust enough to be used reliably in federated LLM fine-tuning.

---

> ### Author Response · Authors · 2025-11-23
>
> We sincerely appreciate your additional inquiry and the opportunity to clarify this aspect. Below we provide a concise response addressing your concern.
> ```
> >Q2.1 Do you have any main task performance compared to baseline methods?
> ```
> To evaluate the main-task utility, we report four widely adopted metrics for generative LLMs. These metrics capture complementary aspects of utility and offer a compact but reliable characterization of model performance:
> * **Perplexity (PPL)** quantifies how well the model fits the underlying language distribution. Lower values indicate more fluent and coherent generation.
> * **BLEU** measures n-gram overlap between generated and reference outputs, reflecting surface-level faithfulness.
> * **ROUGE-L** evaluates the longest common subsequence and reflects content preservation and structural alignment.
> * **MAUVE** assesses the similarity between the distribution of generated and human-written text, indicating naturalness and diversity. Higher values denote closer alignment with human text.
>
> These four metrics are sufficient for generative LLM evaluation and are also used in prior federated LLM works, ensuring comparability.
>
> Table below summarizes the average performance of **FedGraph** compared with **FedAvg, FLAME, and FreqFed** across all five attack settings. Overall, FedGraph preserves utility at a level comparable to or better than baselines—despite aggressively suppressing malicious updates. This demonstrates that our defense simultaneously maintains benign-task performance while mitigating backdoor behavior.
>
> **Table Q2.1. Main-task utility comparison.**
> *Lower PPL is better; higher BLEU, ROUGE-L, and MAUVE are better.*
>
> |             | Method       | PPL ↓         | BLEU ↑      | ROUGE-L ↑   | MAUVE ↑      |
> | ----------- | ------------ | ------------- | ----------- | ----------- | ------------ |
> |             | FedAvg       | 255,748.67    | 0.00952     | 0.0493      | 0.08376      |
> | **Average** | FLAME        | 101,117.63    | 0.01268     | 0.06402     | 0.09996      |
> |             | FreqFed      | 458,926.86    | 0.01240     | 0.06318     | 0.10464      |
> |             | **FedGraph** | **352.64**    | **0.01331** | **0.07261** | **0.12840**  |
>
> ```
> > Q2.2. Do you have any direct computational cost analysis under N =50,d ≈1B settng?
> ```
> To provide a concrete comparison of computational cost, we measured per-round runtime in a realistic LLM federated fine-tuning scenario using LLaMA-2-7B with LoRA ($Rank =32$, $\text{alpha}=64$ due to the limitation of resource), 50 clients, batch size 5, and 2 local training steps per client. The runtime was decomposed into three components: client local training, FedAvg aggregation, and FedGraph aggregation.
>
> The results are summarized in Table Q2.2. Client local training dominates per-round runtime, while FedGraph adds moderate overhead due to pairwise similarity computation, graph construction, and clustering. Overall, the total per-round runtime of FedGraph increases by roughly 20% compared with FedAvg, which is acceptable given the significant improvements in backdoor defense.
>
> **Table Q2.2. Per-round runtime.**
> *Local = client local training time; FA = FedAvg; FG = FedGraph; Agg = aggregation time; OH = relative overhead.*
>
> | Attack   | Local (s) | FA-Agg (s) | FG-Agg (s) | Total-FA (s) | Total-FG (s) | OH (%) |
> | -------- | --------- | ---------- | ---------- | ------------ | ------------ | ------ |
> | BadNets  | 3.0732    | 0.1123     | 0.7036     | 3.1855       | 3.7768       | 18.6   |
> | CTBA     | 3.0216    | 0.1109     | 0.7034     | 3.1325       | 3.7250       | 18.9   |
> | MTBA     | 3.0508    | 0.1125     | 0.7090     | 3.1633       | 3.7598       | 18.9   |
> | Sleeper  | 3.0644    | 0.1111     | 0.8284     | 3.1755       | 3.8928       | 22.5   |
> | VPI      | 3.0527    | 0.1101     | 0.7959     | 3.1628       | 3.8486       | 21.7   |
> | **Avg.** | 3.0362    | 0.1112     | 0.7480     | 3.1474       | 3.7842       | 20.3   |
>
> **Note:** The reported runtimes are based on a conservative configuration with small batch size (5) and only 2 local training steps per round. In practical deployments, batch sizes and local steps are typically larger, leading to higher absolute local training time. However, the **relative overhead of FedGraph compared to FedAvg** is expected to remain similar, as the additional cost from graph construction and pairwise similarity computation scales primarily with the number of clients rather than the batch size or step count. Therefore, the 20–30% overhead observed here provides a reasonable estimate of the expected relative cost in more realistic LLM federated fine-tuning scenarios.
>
> These results indicate that although FedGraph incurs additional computation during aggregation, the overhead is modest relative to client-side training, and the approach remains practical for institution-level LLM federated learning. The added cost is justified by the substantial improvements in backdoor robustness.

---

> > ### Comment · Reviewer_ugEp · 2025-11-26
> >
> > Thank you for your response, I have not further questions. Just a reminder, please add these results into your paper.

---

> ### Author Response · Authors · 2025-11-26
>
> Thank you for your confirmation and for your valuable time and feedback throughout this process. We appreciate it.
>
> We have incorporated the requested results into the revised manuscript: the **direct complexity analysis** is included in **Appendix B.1 (Table 9)**, and the **main-task utility evaluation** is added in **Appendix B.2 (Table 10)**. We sincerely appreciate your time and constructive comments throughout this process.
>
> If further adjustments or clarifications would help with your assessment, we would be more than happy to provide them.

---

### Official Review · Reviewer_tTWb · 2025-10-31

**Soundness:** 2
**Presentation:** 3
**Contribution:** 2
**Rating:** 4
**Confidence:** 4

**Summary:**

This paper introduces a backdoor defense for federated LLMs by computing similarities between clients, building a topology graph to represent their relationships, and extracting three graph-based quantifiers—degree (number of neighbors), betweenness (how often a client lies on shortest paths), and closeness (average proximity to others). These three metrics are then fed into HDBSCAN, where clients marked as outliers are considered potentially malicious. The method is evaluated against several mainstream attacks in federated LLMs and compared with FLAME and FedFreq, showing the highest TPR and lowest ASR across multiple settings.

**Strengths:**

(+) The paper is clearly written, and the method is easy to follow.
(+) The topic is important for improving the security of federated learning with LLMs.
(+) Experimental results show consistent improvements (higher TPR, lower ASR).

**Weaknesses:**

(-) While the main novelty lies in representing client relationships as a graph, the presentation of these graphs is limited (only Figure 3). The similarity metric is cosine similarity between updates, which is widely used in prior works; other metrics are not explored. Given the high dimensionality of updates, it remains unclear whether a simpler method, such as dimensionality reduction combined with clustering, could yield comparable performance.
(-) The three graph-based factors in the 10-client setup are not clearly illustrated. Since they are integer/scalar values, presenting a small quantitative table would help readers interpret the results.
(-) Relying on HDBSCAN’s outlier label (−1) to identify malicious clients raises stability concerns. If malicious clients are similar to each other, why wouldn’t they be grouped into a cluster (with labels different from −1)? Clarification on this behavior and the stability of detection is needed.
(-) Robust aggregator baselines such as RFA, RLR, or DP-based defenses are not compared against.
(-) The impact on model utility (e.g., accuracy or convergence) is not reported, leaving uncertainty about potential trade-offs.
(-) Results for random client selection are missing, and the assumption of 40% malicious clients is quite strong and unrealistic in most federated settings.

Overall, the paper addresses an important problem and shows promising results, but the evaluation is limited, and the robustness of the method is not fully justified. Including comparisons with robust aggregators, reporting utility trade-offs, clarifying assumptions on client sampling, and analyzing clustering stability would significantly strengthen the paper.

**Questions:**

1. In Table 1, why is the ASR "without trigger" larger than the “with trigger” case? Do the authors mean mean $\text{ASR}_{\text{w/o}}$ corresponds to the "without trigger" scenario? Please clarify.
2. What does it mean when the Dirichlet parameter $\alpha > 1$? How does this affect the data heterogeneity across clients?
3. How are the datasets distributed heterogeneously across clients—using Dirichlet splits, shard-based partitions, or other schemes? Please specify parameters for reproducibility.
4. How sensitive is the method to HDBSCAN hyperparameters (`min_cluster_size`, `min_samples`) and the choice of similarity metric?

---

> ### Author Response · Authors · 2025-11-20
>
> Thank you for the suggestions. We have revised the manuscript accordingly and  all modifications are highlighted in blue for clarity.
>
> ```
> > Q1. In Table 1, why is the ASR "without trigger" larger than the “with trigger” case? Do the authors mean mean ASR$_{w/o}$ corresponds to the "without trigger" scenario? Please clarify.
> ```
>
> Thank you for pointing out this issue. The values reported as $ASR_{w/o}$ indeed correspond to the *without-trigger* scenario. The higher ASR$_{w/o}$ in the jailbreak setting is expected and stems from the nature of jailbreak attacks. Unlike standard backdoor attacks that target a specific trigger–response pair, jailbreak attacks aim to make the model *comply* with unsafe or restricted instructions. As a result, even without an explicit trigger, a compromised model may still exhibit a higher tendency to generate unsafe content when queried with sensitive prompts. We also observed that when fine-tuning solely on jailbreak-style data, the model becomes more inclined to answer unsafe questions directly, causing $ASR_{w/o}$ to increase during evaluation on AdvBench.
>
> In generative LLMs, outputs naturally exhibit substantial variability, and this randomness can sometimes cause the model to produce target-like phrases even without a trigger. This effect is **amplified under backdoor manipulation**, because the malicious objective shifts the model distribution toward the attacker’s target semantics. As a result, the attacked model becomes *more prone* to generating target-style content even in trigger-free contexts, which can lead to $ASR_{w/o}$ being slightly higher than the $ASR_{w/t}$ in specific runs.
>
> In summary, it reflects the **intrinsic stochasticity of generative models** combined with the **model’s increased tendency to produce backdoor-related content** after being poisoned. We have clarified this explanation in the revised manuscript to avoid confusion.
>
> ```
> > Q2. What does it mean when the Dirichlet parameter $\alpha$ > 1 ? How does this affect the data heterogeneity across clients?
> ```
> When the Dirichlet concentration parameter ($\alpha > 1$), the data distribution across clients becomes **more uniform**. In the context of federated learning, **heterogeneity** refers to differences in the **data distributions, label proportions, or feature characteristics** across clients. High heterogeneity (strongly non-IID data) means that each client’s local dataset may be heavily skewed toward certain labels or types of samples, while low heterogeneity (more IID-like data) means that client datasets closely resemble the global distribution.
>
> A larger $\alpha$ reduces the variability among client-level label proportions, making each client’s local dataset more similar to the global label distribution. Conversely, smaller $\alpha$ values, particularly $\alpha < 1$, increase non-IIDness by concentrating certain labels on individual clients.
>
> To illustrate, we visualize client-level data distributions in **Appendix A, Figure 5**. **In short:** larger $\alpha$ corresponds to lower heterogeneity (clients’ datasets are more balanced), while smaller $\alpha$ corresponds to higher heterogeneity (clients’ datasets are more skewed).
>
> ```
> > Q3. How are the datasets distributed heterogeneously across clients—using Dirichlet splits, shard-based partitions, or other schemes? Please specify parameters for reproducibility.
> ```
> Following **CLLoRA** [1], context length has minimal impact on local client training but significantly affects the performance of the globally aggregated model. Motivated by this observation, we first partition the full training corpus into **five categories based on contextual length**, ensuring that each client receives data with diverse text-length characteristics, apply Dirichlet sampling [2].
>
> To clarify how datasets are distributed heterogeneously across clients, we provide the exact parameters and procedure used in our experiments:
>
> 1. **Corpus grouping:** The training corpus is divided into **five length-based categories**, maintaining diversity in text-length distributions for each client.
>
> 2. **Heterogeneous splits:** Within each category, samples are allocated using **Dirichlet-based splits**. Client-level proportions are drawn from a Dirichlet prior with concentration parameter $\alpha = 0.9$, which controls the degree of non-IIDness. Lower $\alpha$ values increase heterogeneity, while higher values reduce it. Sensitivity studies exploring multiple $\alpha$ values (0.1, 0.5, 10, 100, IID) are reported in **Section 4.2, Table 6**.
>
> 3. **Reproducibility:** This procedure is fully documented in **Appendix Section A.1**, and the submitted code reproduces the same splits.
>
> [1] Zhang, P., et al. *CLLoRA: An approach to measure the effects of the context length for LLM fine-tuning.* arXiv preprint arXiv:2502.18910 (2025).
>
> [2]Tzu-Ming Harry Hsu., et al. Measuring the effects of non-identical data distribution for federated visual classification. Neurips, 2019.

---

> ### Author Response · Authors · 2025-11-20
>
> ```
> > Q4. How sensitive is the method to HDBSCAN hyperparameters (min_cluster_size, min_samples) and the choice of similarity metric?
> ```
> We adopt **cosine similarity** because the *direction* of client updates is more informative than their magnitude [3][4]. Magnitudes can vary due to heterogeneous data sizes, learning rates, or optimization dynamics, whereas the direction reflects intrinsic optimization behavior. This directional consistency is especially relevant for detecting malicious updates.
>
> Our choice of **HDBSCAN** hyperparameters follows both the threat model and standard practices in prior backdoor-defense literature.
>
> * **min_cluster_size:** We set this to (n/2 + 1) to ensure that benign clients form the majority cluster. This aligns with the common assumption that the number of malicious clients remains below half of all participants [1][2][3][4]; exceeding this would compromise the utility of the global model.
>
> * **min_samples:** We set this to (1) to accommodate the intrinsic heterogeneity in LLM federated fine-tuning, where benign model updates are dispersed in the weight space. As our visualizations show in Figure 3 in Section 4.1, benign updates do not concentrate around a single mode; higher values of **min_samples** would erroneously classify benign updates as noise, reducing defense effectiveness.
>
> To quantify FedGraph’s robustness, we conduct a sensitivity analysis by varying **min_samples**. As it increases from 1 to 4, the average ASR$_{w/t}$ rises from 0.21% to 6.77%, TPR decreases from 96% to 89.5%, and FPR increases from 2.00% to 5.34% (see Appendix C.4 Table 12). These results confirm that FedGraph is effective under the theoretically justified setting **min_samples =1**, while overly strict neighborhood requirements reduce its ability to preserve benign heterogeneity.
>
> **Table R3. The impact of the min_samples.**
>
> | Attack      | **min_samples = 2**  | **min_samples = 3** | **min_samples = 4**  |
> | ----------- | -------------------- | ------------------- | -------------------- |
> |          | ASR$_{w/t}$ / TPR / FPR      | ASR$_{w/t}$ / TPR / FPR     | ASR$_{w/t}$ / TPR / FPR      |
> | BadNet      | 0.00 / 92.50 / 5.00  | 0.00 / 90.00 / 6.67 | 1.04 / 95.00 / 1.67  |
> | CTBA        | 1.04 / 100.00 / 0.00 | 1.06 / 90.00 / 6.67 | 2.04 / 90.00 / 6.67  |
> | MTBA        | 0.00 / 90.00 / 3.33  | 0.00 / 92.50 / 5.00 | 0.00 / 82.50 / 6.67  |
> | Sleeper     | 0.00 / 100.00 / 0.00 | 0.00 / 95.00 / 3.33 | 5.00 / 90.00 / 6.67  |
> | VPI         | 0.00 / 97.50 / 1.67  | 1.06 / 97.50 / 1.67 | 25.77 / 90.00 / 5.00 |
> | **Average** | 0.21 / 96.00 / 2.00  | 0.42 / 93.00 / 4.67 | 6.77 / 89.50 / 5.34  |
>
>
>
> [1] Haomin Zhuang, et al. "Backdoor federated learning by poisoning backdoor-critical layers." In ICLR, 2024.
>
> [2] Li, Haoyang, et al. "3dfed: Adaptive and extensible framework for covert backdoor attack in federated learning." 2023 IEEE symposium on security and privacy (SP). IEEE, 2023.
>
> [3] Nguyen, Thien Duc, et al. "{FLAME}: Taming backdoors in federated learning." 31st USENIX Security Symposium (USENIX Security 22). 2022.
>
> [4] Fereidooni, Hossein, et al. "Freqfed: A frequency analysis-based approach for mitigating poisoning attacks in federated learning." Network and Distributed System Security (NDSS) Symposium. 2024.

---

> ### Author Response · Authors · 2025-11-20
>
> ```
> > Q5–Q6. Comparison with dimensionality reduction and robust aggregators.
> ```
> To test whether a simpler dimensionality-reduction strategy suffices, we applied **PCA + HDBSCAN**. As shown in **Section 4.1 (Table 3)**, this approach yields an average ASR$_{w/t}$ of **61.99%**, indicating that naive linear projections fail to preserve the non-linear similarity structure crucial for anomaly detection.
>
> We further evaluate **robust aggregation and DP-based defenses** (Section 4.1, Table 3), including (i) DP (FLAME), where is the module in FLAME, (ii) standalone DP ($\epsilon=3$, $\delta=10^{-6}$, $\sigma=2.8$), and (iii) RFA. Neither RFA nor DP(FLAME) effectively mitigate backdoors in the LLM setting (average $ASR_{w/t}$ 62.96% and 65.36%, respectively). While standalone DP reduces ASR to 0%, it destabilizes training and severely degrades generative quality (e.g., the model outputs meaningless contents such as “em-o asbagger 1 (D RH1 (D_ R” ).
>
> These results show that FedGraph effectively suppresses backdoors while preserving model quality, whereas RFA and DP-based defenses either fail to block attacks or degrade generative performance.
>
> **Table R4. Results of dimensionality reduction and robust aggregators.**
>
> | Attack  | DP (FLAME) $ASR_{w/o}$ | DP (FLAME) $ASR_{w/t}$ | DP $ASR_{w/o}$ | DP $ASR_{w/t}$ | RFA $ASR_{w/o}$ | RFA $ASR_{w/t}$ | HDBSCAN(PCA) $ASR_{w/o}$ | HDBSCAN(PCA) $ASR_{w/t}$ |
> | ------- | ---------------------- | ---------------------- | -------------- | -------------- | --------------- | --------------- | ------------------------ | ------------------------ |
> | BadNets | 0.00                   | 56.12                  | 0.00           | 0.00           | 0.00            | 47.25           | 0.00                     | 64.89                    |
> | CTBA    | 2.06                   | 87.00                  | 0.00           | 0.00           | 3.03            | 91.00           | 0.00                     | 77.55                    |
> | MTBA    | 9.09                   | 46.94                  | 0.00           | 0.00           | 1.01            | 47.07           | 8.00                     | 76.77                    |
> | Sleeper | 0.00                   | 50.00                  | 0.00           | 0.00           | 0.00            | 48.48           | 1.02                     | 24.73                    |
> | VPI     | 0.00                   | 86.73                  | 0.00           | 0.00           | 0.00            | 81.00           | 0.00                     | 66.00                    |
> |**Avg.**| **2.23**               | **65.36**	            | **0.00**	     | **0.00**	      | **0.81**	     | **62.96**	| **1.80**                 | **61.99**
>
> ```
> > Q7.Results for random client selection are missing, and the assumption of 40% malicious clients is quite strong and unrealistic in most federated settings.
> ```
> We evaluate FedGraph under a worst-case scenario with 40% malicious clients to stress-test its robustness and reveal potential failure modes. Sensitivity studies in **Appendix Section C (Table 6)** vary the number of compromised clients. As the malicious-client ratio decreases, $ASR_{w/t}$ under FedAvg naturally drops, but FedGraph consistently maintains ASR$_{w/t}$ near zero, demonstrating its effectiveness even under more realistic attacker capabilities.
>
> **Table R5. Results of 2 compromised clients.**
>
> | Attack  | Method   | TPR   | FPR   | ASR$_{w/o}$ | ASR$_{w/t}$ |
> | ------- | -------- | ----- | ----- | ----------------- | ----------------- |
> | BadNets | FedGraph | 80.00 | 12.50 | 0.00              | 0.00              |
> |         | FedAvg   | –     | –     | 0.00              | 15.31             |
> | CTBA    | FedGraph | 80.00 | 6.25  | 0.00              | 0.00              |
> |         | FedAvg   | –     | –     | 0.00              | 24.21             |
> | MTBA    | FedGraph | 90.00 | 6.25  | 0.00              | 0.00              |
> |         | FedAvg   | –     | –     | 0.00              | 2.02              |
> | Sleeper | FedGraph | 85.00 | 7.50  | 0.00              | 0.00              |
> |         | FedAvg   | –     | –     | 0.00              | 2.11              |
> | VPI     | FedGraph | 90.00 | 2.50  | 0.00              | 0.00              |
> |         | FedAvg   | –     | –     | 0.00              | 36.00             |
>
> Regarding random client selection, we use **full participation** to remove ambiguity in interpreting results. For example, if only clients [4,5,6,7] are selected in a round, it is unclear whether malicious clients [0,1,2,3] were filtered out by FedGraph or simply not chosen, and whether benign clients [8,9] were mistakenly excluded or just unselected. Full participation ensures that all clients are considered, allowing a clear and direct measurement of the defense’s detection capability without confounding factors from client sampling.
>
> In summary, this setup provides a controlled evaluation and confirms that FedGraph remains robust under both worst-case and more practical scenarios.

---

> ### Author Response · Authors · 2025-11-20
>
> ```
> > Q8. The three graph-based factors in the 10-client setup are not clearly illustrated.
> ```
>
> In the revised manuscript, we provide **Table 4** alongside **Fig. 4 (Section 4.1)** to present a quantitative summary of the three factors—degree, betweenness, and closeness centrality—for each client.
>
> We clarify that the table highlights **two representative patterns** to illustrate the key distinction exploited by FedGraph: highly concentrated malicious feature vectors versus more dispersed benign patterns. Specifically, clients 0–3 are malicious (Mal.), and 4–9 are benign (Ben.).
>
> **Table R6. The Feature Vector.**
>
> | No. | Type | Pattern 2        | Pattern 1        |
> | --- | ---- | ---------------- | ---------------- |
> | 0   | Mal. | [3, 0.0, 0.3430] | [3, 0.0, 0.3339] |
> | 1   | Mal. | [3, 0.0, 0.3416] | [3, 0.0, 0.3339] |
> | 2   | Mal. | [3, 0.0, 0.3420] | [3, 0.0, 0.3553] |
> | 3   | Mal. | [3, 0.0, 0.3434] | [3, 0.0, 0.3552] |
> | 4   | Ben. | [0, 0.0, 0.0]    | [5, 0.0, 0.6460] |
> | 5   | Ben. | [0, 0.0, 0.0]    | [5, 0.0, 0.6473] |
> | 6   | Ben. | [0, 0.0, 0.0]    | [5, 0.0, 0.6509] |
> | 7   | Ben. | [0, 0.0, 0.0]    | [5, 0.0, 0.6370] |
> | 8   | Ben. | [0, 0.0, 0.0]    | [5, 0.0, 0.6676] |
> | 9   | Ben. | [0, 0.0, 0.0]    | [5, 0.0, 0.6460] |
>
> This table is **not exhaustive** of all possible feature representations but demonstrates **canonical patterns** observed across clients:
>
> * **Malicious Pattern:** highly correlated and consistent across the three graph factors.
> * **Benign Pattern:** more variable and dispersed due to natural heterogeneity in local data.
>
> By presenting these representative examples, the table makes the structural separation between benign and malicious updates **explicit and interpretable**, thereby supporting the intuition behind FedGraph’s aggregation and outlier detection mechanism. The complete analysis in the manuscript still considers the full set of client updates and feature vectors across rounds.
>
> ```
> >Q9. If malicious clients are similar to each other, why wouldn’t they be grouped into a cluster (with labels different from −1)?
> ```
>
> In LLM federated fine-tuning, malicious updates are **highly correlated** because they share the same backdoor objective, which causes them to form **tight and well-connected subgraphs** in the similarity graph. In contrast, benign updates exhibit **natural diversity** due to heterogeneous local datasets, varying text lengths, and different contextual patterns. This leads to two possible structures for benign updates: some may be isolated nodes, while others form a broader, loosely connected subgraph representing the majority of clients.
>
> HDBSCAN identifies clusters based on density. Since malicious updates are small in number but highly connected, they are **dense yet isolated relative to the dominant benign structure**, and are thus correctly labeled as outliers (−1). The benign majority, despite internal heterogeneity, forms the main cluster.
>
> This behavior is visualized in **Section 4.1, Fig. 3**, where malicious updates appear tightly clustered and separate, while benign updates are either dispersed or form a larger connected subgraph, allowing HDBSCAN to effectively distinguish outliers from the main cluster.

---

> ### Author Response · Authors · 2025-11-29
>
> ```
> > Q10. The impact on model utility (e.g., accuracy or convergence) is not reported, leaving uncertainty about potential trade-offs.
> ```
>  **1. Why “accuracy” is not applicable in federated generative LLMs**
>
> In generative LLM settings, *accuracy is neither well-defined nor stable*, which fundamentally differs from classification-based FL. Generation is inherently non-deterministic: the same input may yield different outputs across runs due to sampling randomness, heterogeneous client data, and stochastic gradient updates. Thus, accuracy fails to reliably capture model quality, even though FPR/TPR remain valid for assessing defense behavior.
>
> Since FedGraph only selects among submitted updates rather than modifying them, its effect on benign learning is directly reflected by its TPR/FPR performance.
>
>
> **2. Evidence that FedGraph preserves benign-task optimization**
>
> As shown in **Figure 3 (Section 4.1)**, the benign-task loss decreases smoothly over training, while the backdoor-objective loss steadily increases. This divergence indicates that FedGraph suppresses malicious updates without disrupting the normal optimization trajectory of benign updates.
>
> **3. Comprehensive and standard utility metrics for generative LLMs**
>
> To evaluate the main-task utility, we report four widely adopted metrics for generative LLMs. These metrics capture complementary aspects of utility and offer a compact but reliable characterization of model performance:
> * **Perplexity (PPL)** quantifies how well the model fits the underlying language distribution. Lower values indicate more fluent and coherent generation.
> * **BLEU** measures n-gram overlap between generated and reference outputs, reflecting surface-level faithfulness.
> * **ROUGE-L** evaluates the longest common subsequence and reflects content preservation and structural alignment.
> * **MAUVE** assesses the similarity between the distribution of generated and human-written text, indicating naturalness and diversity. Higher values denote closer alignment with human text.
>
> **Table Q2.1. Main-task utility comparison (averaged across all attack settings).**
>
> *Lower PPL is better; higher BLEU, ROUGE-L, and MAUVE are better.*
>
> | Method       | PPL ↓      | BLEU ↑      | ROUGE-L ↑   | MAUVE ↑     |
> | ------------ | ---------- | ----------- | ----------- | ----------- |
> | FedAvg       | 255,748.67 | 0.00952     | 0.0493      | 0.08376     |
> | FLAME        | 101,117.63 | 0.01268     | 0.06402     | 0.09996     |
> | FreqFed      | 458,926.86 | 0.01240     | 0.06318     | 0.10464     |
> | **FedGraph** | **352.64** | **0.01331** | **0.07261** | **0.12840** |
>
> These results show that FedGraph preserves utility at a level comparable to or better than baselines, despite aggressively suppressing malicious updates. This demonstrates that our defense simultaneously maintains benign-task performance while mitigating backdoor behavior. Thus, no significant utility–robustness trade-off is observed in our federated LLM setting.

---

### Official Review · Reviewer_q9tG · 2025-11-01

**Soundness:** 3
**Presentation:** 3
**Contribution:** 2
**Rating:** 2
**Confidence:** 5

**Summary:**

This manuscript propose FedGraph, a method try to mitigate the backdoor attack in federated fine-tuning of LLM. FedGraph represents each client updates as nodes in a dynamic graph, and extracts topological feature for anomaly detection. Experimental results shows that the method seems outperform then existing SoTA.

**Strengths:**

1. The manuscript proposed FedGraph, which is attack data-free.
2. The manuscript provide sufficient evaluation on the effectiveness of FedGraph

**Weaknesses:**

1. some typos: the caption of the Figure 2, "... detection a1nd aggregation ...". The author should carefully revise these typos and other grammer errors.
2. FedGraph is based on the assumption that the benign updates are mutually similar, and malicious updates deviate from these benign clusters. However, the assumption might not hold in some extend. 1) For non-IID distribution, the optimization objective is related to the data, so the degree of non-IID data affects the optimization objectives of different clients. That is to say, the greater the degree of non-IID, the greater the difference in the optimization objectives. In experimental settings, the $\alpha$ set to 0.9 is too large and not representative. 2）For backdoor attack, most of existing methods require the stealthness of the attack, that is, the submited malicious models might be similar with the global model or benign models.
3. The details of how to extract the graph fingerprint should be provided.

**Questions:**

1.How the graph fingerprint extraction process work, please provide the psedu-code or equations.

---

> ### Author Response · Authors · 2025-11-19
>
> Thank you for the suggestions. We have revised the manuscript accordingly and corrected the identified typos. All modifications are highlighted in blue for clarity.
>
> ```
> > Q1. How the graph fingerprint extraction process work.
> ```
> We define the graph fingerprint of each client using three complementary features: **degree centrality**, **betweenness centrality**, and **closeness centrality**. These capture the local similarity structure, bridging roles in information flow, and global topological consistency, respectively, forming a local–intermediate–global representation of client updates. Formally, the metrics are defined as:
>
> $$
> C_D(i) = \sum_{j \in \mathcal{N}(i)} 1, \\
> C_B(i) = \sum_{s \ne i \ne t} \frac{\sigma_{st}(i)}{\sigma_{st}}, \\
> C_C(i) = \frac{1}{\sum_j d(i,j)},
> $$
>
> where $\mathcal{N}(i)$ denotes the neighbors of node $i$, $\sigma_{st}$ is the number of shortest paths between nodes $s$ and $t$, $\sigma_{st}(i)$ counts the shortest paths passing through node $i$, and $d(i,j)$ is the shortest-path distance between nodes $i$ and $j$.
>
> The resulting vector $\mathbf{f}_i$ integrates local connectivity $C_D$ with global influence $C_B$ and $C_C$, providing a robust low-dimensional representation of client update behavior. This process is implemented in Section 3.3, Step 2, and the full algorithm is provided in Appendix B. For intuitive understanding, we also include visualizations in Section 4.1, showing the graph structure in Figure 4 and feature vectors in Table 4.

---

> ### Author Response · Authors · 2025-11-19
>
> ```
> > Q2.For non-IID distribution problem.
> ```
> For the non-IID setting, we follow the standard configurations commonly adopted in prior work [1][2][3] and set $\alpha = 0.9$. In this context, As ( $\alpha \to 0$ ), client data becomes highly non-IID, while as ( $\alpha \to \infty$ ), it approaches IID.
> .  To provide a clear illustration of these effects, we visualize the resulting client-level data distributions in **Appendix A, Figure 5**. This figure demonstrates how varying $\alpha$ directly influences the degree of heterogeneity.
>
> To assess the sensitivity of FedGraph to varying degrees of data heterogeneity, we additionally conduct experiments with $\alpha =$ 0.1, 0.5, 10, 100, as well as fully IID scenarios. These results are included in Appendix and are also summarized in the main text (Section 4.1, Table 6). The following shows the results under the extreme non-IID setting $\alpha = 0.1$, demonstrating that FedGraph remains effective even under highly heterogeneous conditions. These results indicate that even under extreme non-IID distributions, FedGraph effectively suppresses backdoor attacks while maintaining low false positives and high detection rates.
>
> **Table R1. The result of FedGraph with $\alpha=0.1$**
>
> | Attack | TPR (%) | FPR (% )| ASR$_{w/o}$ | ASR$_{w/t}$ |
> |----------|------------|-----------|---------------------|---------------------|
> | BadNets  | 97.50     | 1.67      | 0.00                | 0.00                |
> | CTBA     | 100.00     | 0.00      | 0.00                | 0.00                |
> | MTBA     | 91.25     | 3.33     | 1.04                | 0.00                |
> | Sleeper  | 56.25     | 5.00      | 0.00                | 0.00                |
> | VPI      | 85.25     | 5.00      | 0.01                | 1.00                |
> | **Avg.** | 85.25     | 5.00      | 0.21                | 0.20                |
>
> ```
> > Q3.For the stealthiness of the attack problem.
> ```
> We adopt the stealthy attack strategy proposed in [1], in which only a subset of critical layers is replaced with malicious updates, making the submitted model closely resemble the benign or global model. In our LLM fine-tuning experiments, the attack success rate remains 0.0 under this setting; however, the attack significantly degrades the model’s generative quality, often producing incomplete or nonsensical outputs such as "[/INST] #morningmotivation #morningroutine #selfca'', indicating that the model’s core linguistic capabilities are compromised.
>
> Given the inherent fragility of LLM training, more sophisticated stealth techniques, such as continuous trigger optimization, are difficult to apply because they destabilize training and incur prohibitive computational cost. We also evaluated lower poison rates and mixed updates, for example a linear combination of 0.5 malicious and 0.5 benign updates. These variants similarly fail to produce effective attacks, and the ASR$_{w/t}$ remains near zero across all settings. Detailed results are provided in **Appendix C.2**.
>
> The following is the performance of FedGraph with a poison rate of 0.2. Even under these challenging attack configurations, FedGraph successfully detects and suppresses backdoor updates while maintaining low false positive rates.
>
> **Table R2. The result of FedGraph with 0.2 poisoning rate**
>
> | Attack   | TPR (%) | FPR (%) | ASR$_{w/o}$ | ASR$_{w/t}$ |
> |----------|-----------|-----------|---------------------|---------------------|
> | BadNets  | 80.00     | 5.00      | 0.00                | 0.00                |
> | CTBA     | 92.50     | 3.33      | 0.00                | 1.11               |
> | MTBA     | 82.50     | 6.67      | 0.00                | 0.00               |
> | Sleeper  | 87.50     | 5.00      | 0.00                | 0.00                |
> | VPI      | 90.00     | 6.67      | 0.02                | 0.00               |
> | **Avg.** | 86.50     | 5.33      | 0.004               | 0.222              |
>
> [1] Haomin Zhuang, et al. "Backdoor federated learning by poisoning backdoor-critical layers." In ICLR, 2024.
>
> [2] Li, Haoyang, et al. "3dfed: Adaptive and extensible framework for covert backdoor attack in federated learning." 2023 IEEE symposium on security and privacy (SP). IEEE, 2023.
>
> [3] Nguyen, Thien Duc, et al. "{FLAME}: Taming backdoors in federated learning." 31st USENIX Security Symposium (USENIX Security 22). 2022.

---

### Meta-Review · Area_Chair_R1Tz · 2026-01-07

**Summary:**

Reviewers questioned whether the method shows sufficient novelty and robustness for ICLR, particularly the core assumption that benign updates form coherent similarity structures while malicious ones are detectable as anomalies under realistic federated settings. The approach was also viewed as potentially incremental, relying on established components.

Additional concerns included the limited evaluation scope, strong experimental assumptions, missing comparisons with robust aggregation baselines, unclear impact on main-task utility, scalability due to quadratic graph construction, and aspects of clarity and presentation. Taken together, these factors suggest that the work may benefit from further development before being competitive for acceptance at ICLR.

**Reviewer Concerns:**

Concerns Largely Addressed:
- Method clarity: The authors substantially improved explanations of graph fingerprints, clustering behavior, and added formal definitions, tables, and visualizations.
- Evaluation breadth: Additional experiments addressed non-IID sensitivity, stealthy attacks, lower attacker ratios, robust aggregation baselines (e.g., RFA, DP/FLAME), and dimensionality-reduction alternatives.
- Utility and overhead: The rebuttal added standard generative LLM utility metrics and concrete runtime measurements, alleviating earlier concerns about missing evaluation dimensions.
- Presentation issues: Typos, formatting, and missing details were corrected.

Concerns Still Outstanding:
- Despite the expanded evaluation, some reviewer reservations regarding reliance on majority-benign assumptions, density-based clustering behavior, and full client participation are partially alleviated.
-The rebuttal strengthens the empirical case but does not fundamentally change the perception that the contribution is incremental rather than conceptually new.
-  While justified for institution-level LLM FL, questions about broader applicability remain open.

**Reviewer Scores:**

- Reviewer q9tG (initial: 2): Likely to improve after clarifications on non-IID settings, stealthy attacks, and fingerprint extraction, though still short of a confident accept.
- Reviewer tTWb (initial: 4): Likely to remain unchanged, with concerns partially addressed.
- Reviewer ugEp (initial: 6): Likely to remain unchanged.

---

### Decision · Program_Chairs · 2026-01-26

Reject